



# Earthquake hazard characterization by using entropy: application to northern Chilean earthquakes

Antonio Posadas[1, 2], Denisse Pasten [3], Eugenio E. Vogel [4,5], Gonzalo Saravia[6]

[1]Departamento de Química y Física, Universidad de Almería, 04120 Almería, Spain.
[2]Instituto Andaluz de Geofísica, Campus Universitario de Cartuja, Universidad de Granada, 18071 Granada, Spain
[3]Departamento de Física, Facultad de Ciencias, Universidad de Chile, Santiago, Chile
[4]Departamento de Ciencias Físicas, Universidad de La Frontera, Casilla 54-D, Temuco, Chile
[5]Center for Nanoscience and Nanotechnology (CEDENNA), Santiago, Chile
[6]Los Eucaliptus 1189, Temuco 4812537, Chile

*Correspondence to*: Antonio Posadas (aposadas@ual.es)

**Abstract.** The mechanical description of the seismic cycle has an energetic analogy in terms of statistical physics and the Second Law of Thermodynamics. In this context, an

earthquake can be considered as a phase transition, where continuous reorganization of stresses and forces reflects an evolution from equilibrium to non-equilibrium states and we can use this analogy to characterize the earthquake hazard of a region. In this study, we used 8 years (2007–2014) of high-quality Integrated Plate Boundary Observatory Chile (IPOC) seismic data for >100,000 earthquakes in northern Chile to test the theory

that Shannon entropy, $H$, is an indicator of the equilibrium state of a seismically active region. We confirmed increasing $H$ reflects the irreversible transition of a system and is linked to the occurrence of large earthquakes. Using variation in $H$, we could detect major earthquakes and their foreshocks and aftershocks, including 2007 $M_W$ 7.8 Tocopilla earthquake, 2014 $M_W$ 8.1 Iquique earthquake, and the 2010 and 2011 Calama

earthquakes ($M_W$ 6.6 and 6.8, respectively). Moreover, we identified possible periodic seismic behaviour between 80 and 160 km depth.

## 1 Introduction

The seismicity of a region contains abundant information that can be used, from different
points of view, attempting to know when an earthquake is going to occur. In physics, Entropy is one of the most fascinating, abstract and complex concepts. The present paper shows how to use Entropy to characterize the occurrence of earthquakes, i.e. to have a characterisation of the seismic hazard in entropic terms.

It is well known (e.g. Nikulov, 2022) that the second law of thermodynamics postulates
the existence of irreversible processes in physics: the total entropy of an isolated system can increase, but cannot decrease. Namely, only those phenomena for which the entropy of the universe increases are allowed. Thus, in seismology, it is natural to use entropy to find out future states that a region of the Earth's crust can access from its current state (Akopian, 2015).



The concept of entropy and its connection to the Second Law of Thermodynamics was proposed by Clausius in 1865 (Clausius, 1865) and a few years later, Boltzmann realised that entropy could be used to connect the microscopic motion of particles to the macroscopic world; in his analysis, entropy is proportional to the number of accessible micro-states of the system ($\Omega$) and is expressed by the famous Boltzmann equation:

$$S = kln\,\Omega \,, \tag{1}$$

where $k$ is Boltzmann's constant. Ben-Naim (2020) stated that, at first glance, Boltzmann's entropy and Clausius' entropy are absolutely different; however, there is complete agreement in calculating changes in entropy using the two methods (up to a multiplicative constant). The generalization of Boltzmann's entropy for systems described by other macroscopic variables corresponded to Gibbs (Zupanovic and Domagoj, 2018) and can be written as:

$$S = -k \sum_{i=1}^{\Omega} p_i \log p_i \,, \tag{2}$$

where $p_i$ is the probability of the system being in the *i-th* state. Shannon (1948) and Shannon and Weaver (1949) introduced Boltzmann-Gibbs's entropy concept into communication theory and defined the measure of information as:

$$I(p) = \sum_{i=1}^{\Omega} p_i \log p_i \,, \tag{3}$$

where $p$ is the distribution of states and $p_i$ is the relative frequency for each event $i$. The function $I(p)$ is called 'Shannon information' because it is a measure of knowledge; therefore, $-I(p)$ denotes a lack of knowledge or ignorance as Majewski (2001) has highlighted. Clearly, $I(p)$ is always negative or zero; as such, it is possible to define the 'Shannon information entropy' ($H$) as the negative information measure (Ben-Naim, 2017); that is:

$$H(p) = -I(p) = - \sum_{i=1}^{W} p_i \, log \, p_i \,, \tag{4}$$

which is always positive or zero. Some (relatively) recent research carried out in the field of information theory suggests that the above expressions can be generalised. Thus, Tsallis (1988) proposed the use of:

$$S_\tau = \frac{k}{\tau - 1} \left( 1 - \sum_{i=1}^{W} p_i^\tau \right) \,, \tag{5}$$

where $\tau$ is a real number called the entropic index. The standard distribution that characterises Boltzmann-Gibbs statistics is a particular case of Tsallis entropy in the limit




of τ = 1. Others generalizations, such as Renyi entropy, can be found in the scientific literature (e.g. Majewski and Teisseyre, 1997).

From the point of view of classical thermodynamics (Varotsos *et al.*, 2011; Vargas *et al.*, 2015; Sarlis *et al.,* 2018; Vogel *et al.*, 2020; Telesca *et al.*, 2022, Varotsos *et al.*, 2022), but also statistical mechanics (Michas *et al.*, 2013; Vallianatos *et al.*, 2015; Papadakis *et*
*al.*, 2015; Vallianatos *et al.*, 2016; Vallianatos *et al.*, 2018), variation in Entropy has been widely used in seismology as an indicator of the evolution of a system (from precursor papers such as Rundle *et al.*, 2003 or Sornette and Werner, 2009, to recent ones from Posadas *et al.*, 2022, Pasten *et al.*, 2022 or Posadas and Sotolongo, 2023).

In this paper, we used 8 years (2007–2014) of high-quality Integrated Plate Boundary
Observatory Chile (IPOC) seismic data for >100,000 earthquakes in northern Chile to test the theory that Shannon entropy, $H$, is an indicator of the equilibrium state of a seismically active region. Moreover, we will rough out a thermodynamics vision of the seismic cycle to characterize the seismic hazard of the northern Chilean seismicity.

**2 Methods**

**2.1 Theoretical framework**

Let us start with a representation of the state of a given seismically active region from the distribution of earthquakes with magnitudes $M$ associated with time $t$; that is, $P(M)$. Thus, entropy, $H$, postulated by Shannon, which is associated with information flow, can be
reformulated (De Santis *et al.*, 2019) as:

$$H(t) = -\int_{M_0}^{\infty} P(M) \times log(P(M)) dM \tag{6}$$

where $M_0$ is the threshold magnitude (i.e., the magnitude for which the seismic catalogue is complete). There are two restrictive conditions to solve that integral. First:

$$\int_{M_0}^{\infty} P(M) dM = 1 \tag{7}$$

The second arises from the fact that the average value of all possible magnitudes $\bar{M}$, in a certain period, is:

$$\bar{M} = \int_{M_0}^{\infty} M \times P(M) dM \tag{8}$$

The Second Law of Thermodynamics requires that there exists a distribution under which $H$ would be at its maximum value while under the two restrictive conditions; that is, the spontaneous development of the system from a state of non-equilibrium to a state of equilibrium is a process in which entropy increases and the final state of equilibrium





corresponds to the maximum entropy. Thus, the problem can be solved by applying the
Lagrange multiplier method; to do that, we define the lagrangian $\mathcal{L}$ as:

$$\mathcal{L}\left(P(M)\right) = H(P(M)) - \lambda_1 \int_{M_0}^{\infty} P(M)dM - \lambda_2 \int_{M_0}^{\infty} M\, P(M)\, dM \qquad (9)$$

where $\lambda_1$ and $\lambda_2$ are Lagrange's multipliers; then, it is possible to deduce the probability
density function in the form (Feng and Luo, 2009):

$$P(M) = \frac{1}{\overline{M} - M_0} \exp\left(-\frac{M - M_0}{\overline{M} - M_0}\right) \qquad (10)$$

On the other hand, if we have $N$ earthquakes and $n$ denotes the number of earthquakes
with magnitude $M$:

$$P(M) = \frac{n}{N} \qquad (11)$$

then, we match both formulas and take logarithms to get:

$$\log n = \log\left(\frac{N}{\overline{M} - M_0}\right) + \frac{M_0 \cdot \log(e)}{\overline{M} - M_0} - \frac{\log(e)}{\overline{M} - M_0} \cdot M \qquad (12)$$

But, the Gutenberg-Richter relationship (Gutenberg and Richter, 1944) states that the
distribution of earthquake magnitudes follows an empirical and universal relationship:

$$\log n = a - bM \qquad (13)$$

where $n$ is the cumulative number of earthquakes with a magnitude equal to or larger than
$M$, and $a$ and $b$ are real constants that may vary in space and time. Parameter $a$
characterises the general level of seismicity in a given area during the study period (i.e.,
the higher the $a$ value, the higher the seismicity), whereas parameter $b$, which is typically
close to 1, describes the relative abundance of large to smaller shocks. Now, identifying
terms from Eqs. 12 and 13, we obtain:

$$a = \log\left(\frac{N}{\overline{M} - M_0}\right) + \frac{M_0 \cdot \log(e)}{\overline{M} - M_0} \qquad (14)$$

and




$$b = \frac{\log(e)}{\overline{M} - M_0} \qquad (15)$$

Hence, the probability density function (Eq. 10) can be rewritten as:

$$P(M) = \frac{b}{\log(e)} \cdot 10^{-b(M - M_0)} \qquad (16)$$

and, finally, substituting into Eq. 6, we get (Posadas *et al.*, 2022):

$$H = -\int_{M_0}^{\infty} \frac{b \cdot 10^{-b(M - M_0)}}{\log(e)} \cdot \log\left(\frac{b \cdot 10^{-b(M - M_0)}}{log(e)}\right) dM =$$

$$= -\log(b) + \log(e \cdot log(e)) \qquad (17)$$

After computing $b$ from the classical Utsu expression (Utsu, 1965):

$$b = \frac{\log(e)}{\overline{M} - (M_0 - \frac{\Delta M}{2})} \qquad (18)$$

where $\Delta M$ is the resolution of magnitude (usually $\Delta M = 0.1$), the value of entropy can be found.

## 2.2 Methodology

Our analysis approach included three steps:

1. First, the value of the threshold magnitude ($M_0$) is a critical choice. There are two main classes of methods to evaluate $M_0$: catalogue-based methods (e.g., Amorèse, 2007) and network-based methods (e.g., D'Alessandro *et al.*, 2011). We used a catalogue-based method because the necessary inputs were available from our dataset. Although some studies estimate the value of $M_0$ by fitting the linear Gutenberg–Richter relationship to the observed frequency–magnitude distribution (the magnitude at which the lower end of the frequency–magnitude distribution departs from the Gutenberg–Richter relationship is taken as $M_0$ (Zúñiga and Wyss, 1995)), several other methods can better determine the threshold magnitude. Catalogue-based techniques include day-to-night noise modulation (day/night method) (Rydele and Sacks, 1989), the Entire Magnitude Range (Ogata and Katsura, 1993), the MAXC technique or Goodness-of-Fit Test (GFT) (Wiemer and Wyss, 2000), b-value stability (MBS) (Cao and Gao, 2002), and median-based analysis of the segment slope (MBASS) (Amorèse, 2007). The MAXC technique is mainly used in applied techniques and was chosen here; however, the results do not differ significantly among these approaches.


2. Second, the time interval *W* was determined for the calculation of entropy, using the minimum number of earthquakes to calculate *H*. The time interval can be chosen by defining a cumulative, moving, or overlapping earthquake window. Here, the results are presented for a sliding window to avoid the memory effect. It turns out that the results are substantially the same regardless of the approach taken. On the whole, the final window size offered a reasonable compromise between resolution and smoothing. The width of the window was chosen by following the approach of De Santis *et al.* (2011), which is based on meaningful values of *b*. In short, 200 events is the minimum needed to perform a robust statistical estimation of *b* and *H*. This has been confirmed by previous statistical analyses of *a* and *b* values (Utsu, 1999). However, larger values of *W* can be adopted depending on the relative error when entropy is computed (Posadas *et al.*, 2022); this criterion is explained below in the Results section.

3. Finally, the entropy function was obtained for each time *t* following Eq. 17. By convention, the time attributed to each point of the analyses was the time of the last seismic event considered in each window. The occurrence of a large earthquake (or the accumulation of several important ones) is expected to lead the seismic system to a state of greater disorder. Then, any earthquake is an irreversible transition to a new state carrying an increase in entropy. Once the major shock is over, entropy returns to stable values.

**3 Data: the northern Chilean seismicity**

The Pacific Ring of Fire, a 40,000 km horseshoe marking the tectonic boundaries of the Pacific Ocean (primarily along the boundaries of the Pacific Plate), hosts 90% of Earth's seismic activity and 75% of the active volcanoes. Also known as the Circum-Pacific Belt, it extends from Tonga and the New Hebrides islands through Indonesia, the Philippines, Japan, the Kuril and the Aleutian Islands, to the western coast of North America, before ending in the Cordillera de los Andes of South America. Among these regions, the Northern Chile Forearc experiences abundant interplate and intraplate earthquakes, intermediate and deep earthquakes associated with subduction, and a high tsunami risk along coastal areas. Events such as 2007 $M_W$ 7.8 Tocopilla earthquake (Delouis *et al.*, 2009), 2010 $M_W$ 8.8 Maule megathrust earthquake (Derode *et al.*, 2021), and 2014 $M_W$ 8.1 Iquique earthquake (Cesca *et al.*, 2016) highlight the special relevance of this region. As such, monitoring seismic and volcanic activity in northern Chile using dense seismic networks (permanent and temporary) to create extensive high-quality seismic catalogues is a priority. To this end, the Integrated Plate Boundary Observatory Chile (IPOC), established by a network of European and South American institutions, operates a wide system of instruments and projects dedicated to the study of earthquakes and deformation at the continental margin of Chile (https://www.ipoc-network.org/). The network extends from the Peru–Chile border in the north to the city of Antofagasta in the south, and from the coast in the west to the high Andes in the east.





**Table 1.** Earthquakes with magnitudes of > 6.5 in the Integrated Plate Boundary Observatory Chile (IPOC) catalogue for the period 2007 to 2014.

| Date (yyyy/mm/dd) | Time | Latitude | Longitude | Depth (km) | $M_W$ | Name |
|---|---|---|---|---|---|---|
| 2007/11/14 | 15:40:50 | −22,332 | −70,044 | 49.24 | 7.8 | Tocopilla earthquake |
| 2007/12/16 | 08:09:13 | −23,298 | −70,379 | 64.22 | 6.9 | Aftershock of Tocopilla earthquake |
| 2010/03/04 | 22:39:24 | −22,391 | −68,572 | 109.51 | 6.6 | Calama 2010 earthquake |
| 2011/06/20 | 16:35:58 | −21,894 | −68,554 | 132.84 | 6.8 | Calama 2011 earthquake |
| 2014/03/16 | 21:16:28 | −19,955 | −70,860 | 17.86 | 6.6 | Foreshock of Iquique earthquake |
| 2014/04/01 | 23:46:46 | −19,589 | −70,940 | 19.91 | 8.1 | Iquique earthquake |
| 2014/04/03 | 02:43:14 | −20,595 | −70,585 | 21.96 | 7.6 | Aftershock of Iquique earthquake |


In this study, we used high-quality IPOC data from 2007 to 2014 (the period for which data are publicly available) to test the theory that Shannon entropy (we will use Shannon entropy but whatever other such as Tsallis entropy, e.g. Vallianatos *et al.*, 2015, Vallianatos *et al.*, 2018, Khordad *et al.*, 2022 or Rastegar *et al.*, 2022 could be adopted)

represents an indicator of the equilibrium state of a seismically active region (or seismic system); we hypothesized that the relationship between increasing entropy and the occurrence of large earthquakes reflects the irreversible transition of a system. The data included records of 101,601 accurately located earthquakes within an epicentral area of 17ºS–25ºS and 66ºW–72ºW (Figure 1). A comprehensive study of the dataset can be

found in Sippl *et al.* (2018).


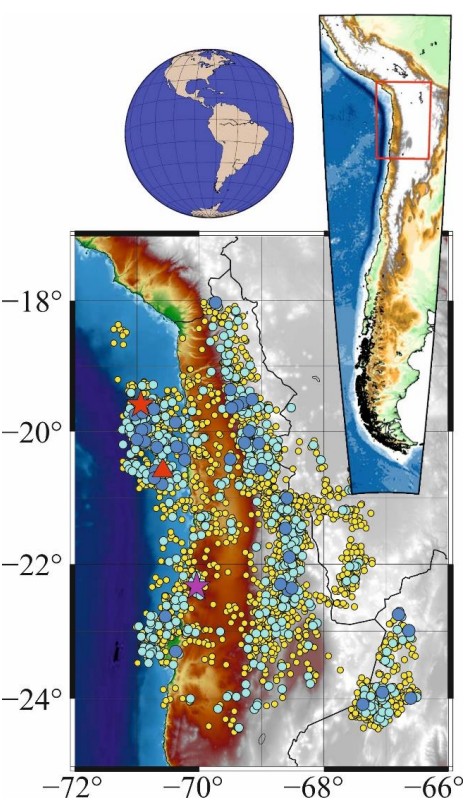

**Figure 1.** Seismicity within an epicentral area of 17°S–25°S and 66°W–72°W between 2007 and 2014. Data are from the Integrated Plate Boundary Observatory Chile (IPOC) catalogue, which contains > 100,000 earthquakes; however, only events with magnitudes of > 4.0 are shown here (3,960 events in total). Circle colours denote event magnitudes: yellow = 4.0–4.9, cyan = 5.0–5.9, and blue = 6.0–6.9. Earthquakes with magnitudes of > 7.0 include 2007 $M_W$ 7.8 Tocopilla earthquake (magenta star), 2014 $M_W$ 8.1 Iquique earthquake (red star), and its main aftershock ($M_W$ = 7.6, shown by the red triangle).

## 4 Results

The seismic catalogue contains 32 earthquakes with magnitudes of 6.0 or greater, 7 of which have magnitudes of > 6.5 (Table 1). The two largest earthquakes are the $M_W$ 7.8
Tocopilla earthquake (November 14, 2007) and $M_W$ 8.1 Iquique earthquake (April 1, 2014). Figure 2 shows a time series of events for earthquakes with magnitudes of > 4.0; the number of earthquakes versus time is shown in Figure 3.

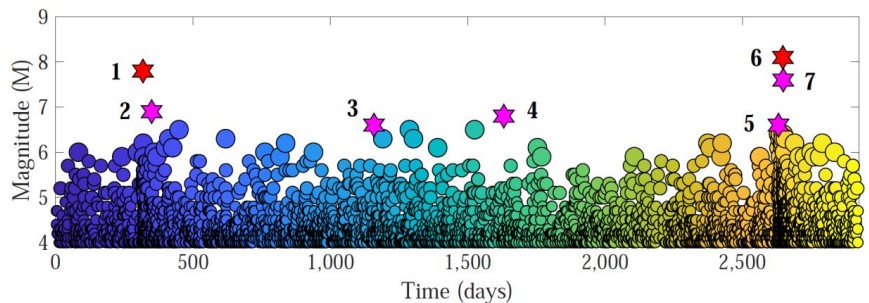

**Figure 2.** Magnitude versus time for earthquakes with magnitudes of > 4.0 within an epicentral area of 17ºS–25ºS and 66ºW–72ºW. Stars correspond to the earthquakes listed in Table 1, including the (1) 2007 $M_W$ 7.8 Tocopilla earthquake, (2) 2007 $M_W$ 6.9 Tocopilla aftershock, (3) 2010 $M_W$ 6.6 Calama earthquake, (4) 2011 $M_W$ 6.8 Calama earthquake, (5) $M_W$ 6.6 foreshock of the Iquique earthquake, (6) $M_W$ 8.1 Iquique earthquake, and (7) $M_W$ 7.6 aftershock of the Iquique earthquake. Circles' size increases gradually with magnitude and colour, from blue to yellow, highlighting the temporal evolution.

.

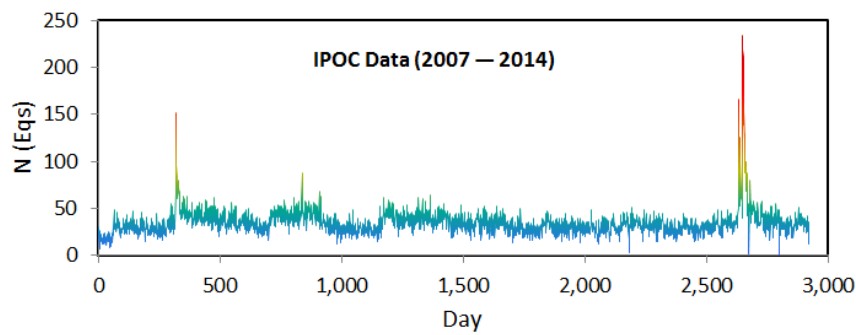

**Figure 3.** Number of daily earthquakes from 2007 to 2014 within an epicentral area of 17ºS–25ºS and 66ºW–72ºW. The seismic crises associated with the 2007 $M_w$ 7.8 Tocopilla earthquake and 2014 $M_w$ 8.1 Iquique earthquakes are clearly distinguished by the two prominent peaks.


First, the threshold magnitude $M_0$ is needed; to get it, we used the MAXC technique as we have mentioned before. Then, the Gutenberg-Richter relationship was got (Figure 4) and a value of $M_0 = 2.2$ is found.

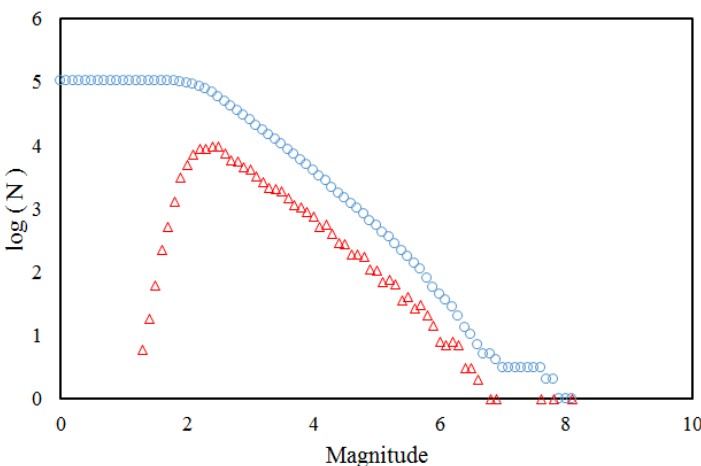

**Figure 4.** Gutenberg–Richter Law for the Integrated Plate Boundary Observatory Chile (IPOC) catalogue from 2007 to 2014 within an epicentral area of 17ºS–25ºS and 66ºW–72ºW. Blue circles denote the cumulative number of earthquakes; red triangles denote the non-cumulative number of earthquakes. Based on the maximum curvature (MAXC) technique (Wiemer and Wyss, 2000), $M_0 = 2.2$.

The second step of our method is to determine the width of window *W* for the windowing process. Figure 5 shows the relative error of entropy versus window width. The choice of *W* must consider that values of *b* should be significant. One way to objectify this choice of *W* is to study the relative error when obtaining the entropy. Utsu's formalism (Utsu 1965) showed that the uncertainty associated with *b* value, interpreted as the error in the 205  *b* value determination, is given by:

$$\sigma = \frac{b}{\sqrt{N}} \tag{19}$$

From the expressions 17 and 19, it is easy to get that, for an entropy value *H*, the error margins are:

$$\Delta H = \log\left(\frac{b + \Delta b}{b - \Delta b}\right) \tag{20}$$

Hence, the relative error can be calculated as:

$$\varepsilon\,(\%) = \frac{100}{H} \cdot log\left(\frac{b + \Delta b}{b - \Delta b}\right) \tag{21}$$


From Figure 5, as the window width increases, the error decreases; when the window width is 4,000 earthquakes (blue line), the error is barely 1%. Overall, the relative errors




of entropy range between 0.5% and 2% for window widths of $> 500$ cumulative earthquakes. From this point of view, the choice of $W$ must be a reasonable compromise between calculated errors and the visibility of the results. We ultimately chose a window of $W = 3,000$ earthquakes (yellow line), for which the relative error of entropy is close to 1% and remains practically constant.

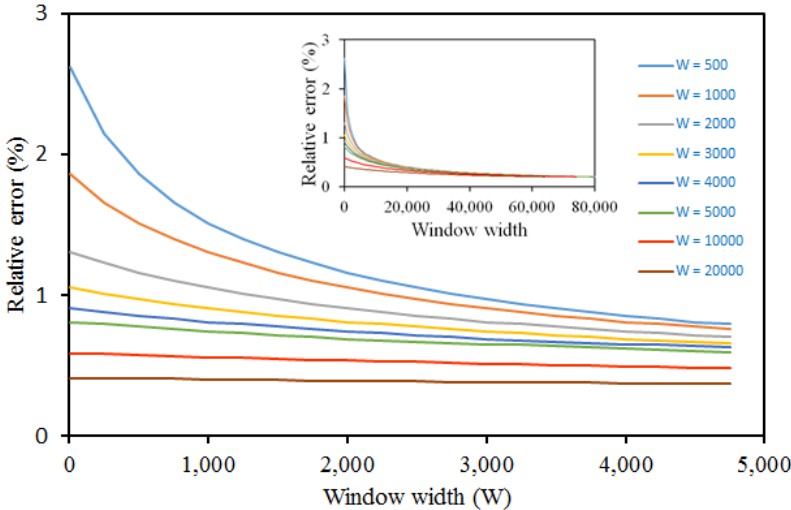

**Figure 5.** Relative error as a function of the given initial window width. For example, the cyan line corresponds to an initial window width of W = 500, for which the calculated relative error in entropy is 2.7%.

The threshold magnitude and width of the window for the windowing process have been set to $M_0 = 2.2$ and $W = 3,000$, respectively; this reduced the size of the catalogue to 84,593 events. Finally, the third step is to get Entropy $H$. The evolution of entropy with time from the windowing process is shown in Figure 6. Sudden changes in entropy are evident and correspond to the times of the largest earthquakes. Levels of change in the absolute values of entropy increase with increasing earthquake magnitude. The entropy change for the Tocopilla earthquake reached $H = 0.35$, while for the Calama 2010 and 2011 earthquakes, it barely exceeded $H = 0.25$. For the Iquique earthquake and its large foreshock and aftershock, the entropy value reached $H = 0.45$.

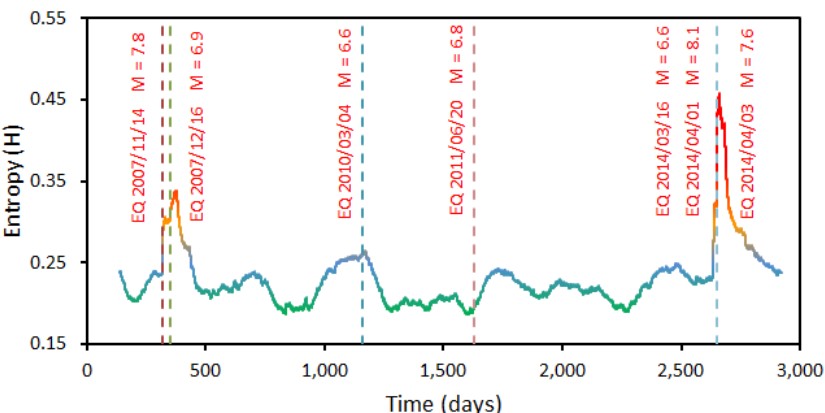

**Figure 6.** Time series of Shannon entropy, H, with the occurrence times of $M_W >$ 6.5 earthquakes shown by dashed lines (note that the large foreshock, mainshock, and large aftershock of the Iquique earthquake occurred close together in time; as such, only a single dashed line is shown). Sudden changes in entropy are clearly identifiable and coincident with large earthquakes.

Chilean seismicity is not only shallow seismicity; in fact, deep abundant earthquakes occur as correspond to a subduction region; then, we also investigated entropy variation as a function of earthquake type, as defined by depth (Figures 7 and 8), as follows. Zone
A: intraplate earthquakes characterised by shallow depth (0–80 km) and a tectonic origin. Zone B: interplate earthquakes characterised by intermediate depth (80–160 km) and related to the contact between the two plates. Zone C: slab earthquakes that occur at large depths (> 160 km) in the slab of the underlying plate.

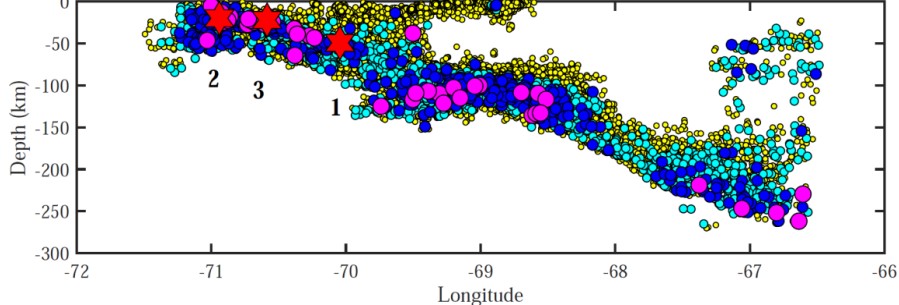

**Figure 7.** Earthquake depth versus longitude for earthquakes with magnitudes of > 2.0. Circle colours denote event magnitudes: yellow = 2.0–3.9, cyan = 4.0–4.9, blue = 5.0–5.9, and magenta = 6.0–6.9. Red stars denote earthquakes with magnitudes of > 7.0, including the (1) 2007 $M_W$ 7.8 Tocopilla earthquake, (2) 2014 $M_W$ 8.1 Iquique earthquake, and (3) 2014 $M_W$ 7.6 aftershock of the Iquique earthquake.



The analysis of threshold magnitudes for zones A, B, and C, as well as the calculation of
window $W$ were as described above for the previous calculation of $H$ (see Figure 9 for
epicentral maps of the three zones and the computation of $M_0$ in each). Figure 10 shows
the time series of entropy for each of the three zones. In zone A, sudden changes in
entropy were coincident with the Tocopilla and Iquique earthquakes. Zones B and C show
low-amplitude sawtooth fluctuations in entropy (maximum $\Delta H$ of $\leq 0.09$ vs. $\Delta H \approx 0.5$ in

zone A). The entropy variations in zones B and C are negligible compared with those in
zone A.

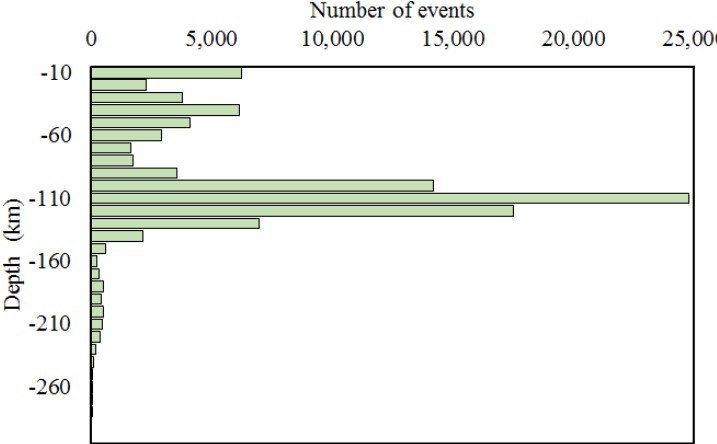

**Figure 8.** Histogram of earthquake depth from 2007 to 2014 within an epicentral area of 17ºS–
25ºS and 66ºW–72ºW. Bins have a 10 km resolution and three regions can be differentiated:
zone A (up to 80 km depth), zone B (80–160 km depth), and zone C (> 160 km depth).

In zone B (Figure 11), the 2010 and 2011 Calama earthquakes ($M_W$ 6.6 and $M_W$ 6.8 events
on days 1,158 and 1,631, corresponding to April 4, 2010 and June 20, 2011, respectively)

are clearly identifiable by increases in entropy. Other peaks before and after these
earthquakes are coincident with either smaller earthquakes or clusters of smaller
earthquakes ($M_W$ 5.5–6.5), including a $M_W$ 6.5 event on March 24, 2008 (day 448); a
group of earthquakes between December 4, 2008 and March 27, 2009 (days 703–816,
magnitudes of 5.8–6.0), a $M_W$ 5.9 earthquake on August 8, 2012 (day 2,107); a cluster of

earthquakes between July 10, 2013 and January 7, 2014 (days 2,382–2,563, magnitudes
of 5.9–6.2); and, two earthquakes on March 31 and August 23, 2014, both with
magnitudes of 6.2 (days 2,646 and 2,791, respectively).

The apparent periodicity in zone B suggests carrying out a Fourier analysis of the entropic
signal. The entropic signal is not uniformly sampled in the time domain; for this reason,

it was averaged to the tenth part of the day and, subsequently, an interpolation was made
for points with no sample. Thus, the resulting entropic signal was uniformly sampled and
a fast Fourier transform was feasible.

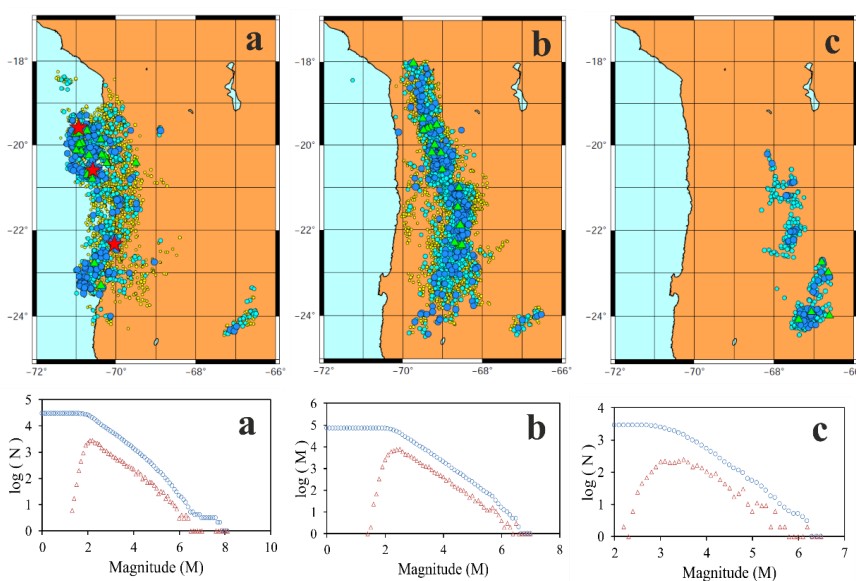

**Figure 9.** Epicentrally represented earthquake activity and non-cumulative and cumulative Gutenberg–Richter relationships in zones A–C for earthquakes with magnitudes of > 3.0. (a) Zone A (0–80 km), (b), zone C (80–160 km), and (c) zone C (>160 km). Symbol colours denote earthquake magnitude: yellow circles = 3.0–3.9, cyan circles = 4.0–4.9, blue circles = 5.0–5.9, green triangles = 6.0–6.9, and red stars = > 7.0. Based on the maximum curvature (MAXC) technique (Wiemer and Wyss, 2000), $M_0$= 2.2 in zones $A$ and B, and 3.2 in zone C.

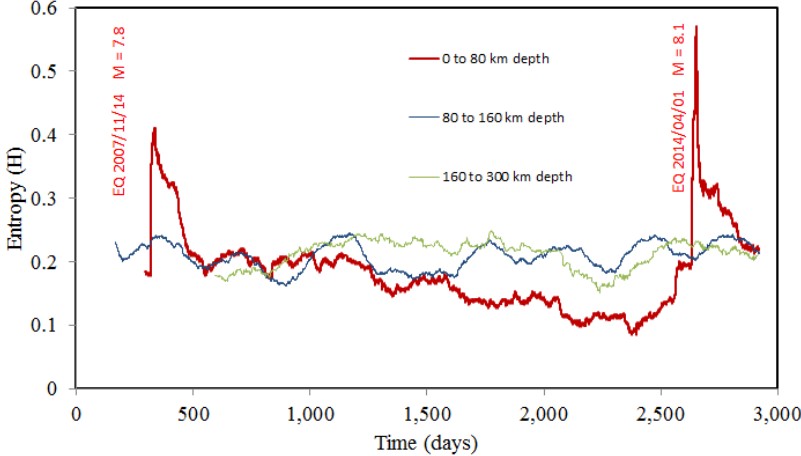

**Figure 10.** Time series of Shannon entropy, H, within different depth intervals. The red line denotes zone A (earthquakes with depths of 0–80 km), the blue line denotes zone B (80–160 km), and the green line denotes zone C (> 160 km). The 2007 $M_W$ 7.8 Tocopilla earthquake and 2014 $M_W$ 8.1 Iquique earthquake are marked and are coincident with increases in entropy in zone A.

The Fourier transform of the entropic signal (Figure 12) revealed that the peaks of the predominant amplitude have frequencies of 0.00048 and 0.00119 $days^{-1}$, corresponding to periods of ~2,100 and 840 days, respectively. The 840-day period approximately reproduces the sequence of M > 5.5 earthquakes. For instance, 840 days after the Tocopilla earthquake (November 14, 2007) was March 3, 2010, which is 1 day before the 2010 Calama 2010. However, given the relatively short period covered by the data (8 years), this Fourier analysis is necessarily preliminary. Further studies with observation periods from 2015 until the present are needed to confirm these results.

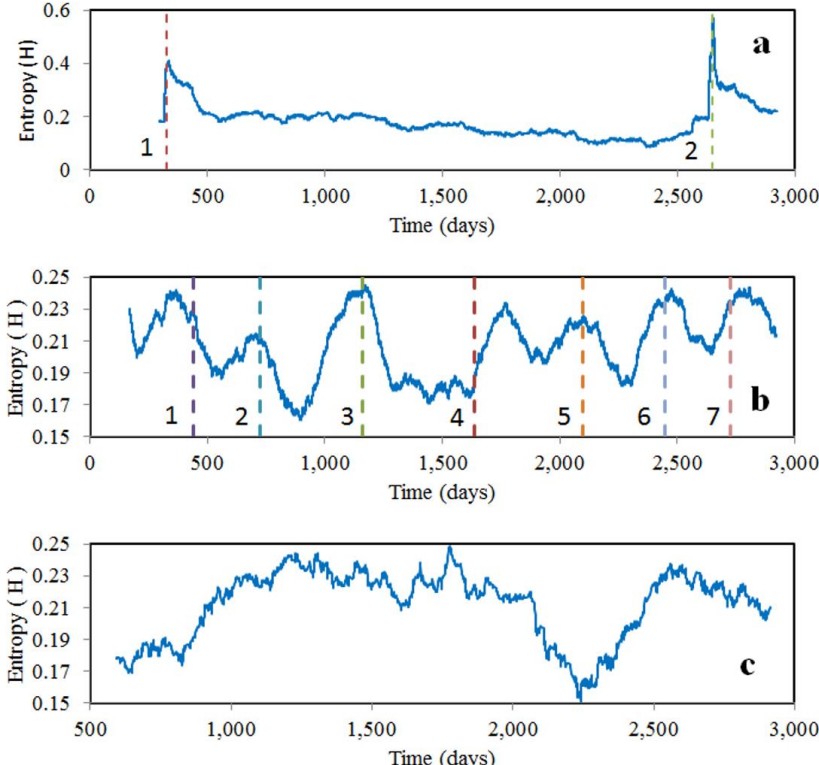

**Figure 11.** Time series of Shannon entropy, H, within different depth intervals. (a) Zone A (earthquakes with depths of 0–80 km), (b) zone B (80–160 km), and (c) zone C (> 160 km). The relative change in entropy in zone A is ~0.5 units compared with 0.09 units in zones B and C. Lines 1 and 2 in (a) correspond to the 2007 $M_W$ 7.8 Tocopilla earthquake and $M_W$ 8.1 Iquique earthquake, respectively; lines 1 to 7 in (b) correspond to the $M_W$ 6.5 March 2008 earthquake, clusters of earthquakes with magnitudes ranging from 5.8 to 6.0 from December 2008 to March 2009, the 2010 $M_W$ 6.6 Calama earthquake, the 2011 $M_W$ 6.8 Calama earthquake, the 2012 $M_W$ 5.9 earthquake, clusters of earthquakes with magnitudes ranging from 5.9 to 6.2 from July 2013 to January 201, and the two 2014 $M_W$ 6.2 earthquakes.
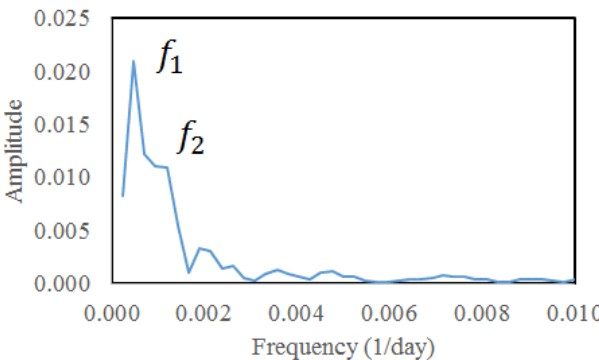

**Figure 12.** Spectrum for the entropic signal of zone B (80–160 km). The two peak amplitudes have frequencies of $f_1 = 0.00048 \, \text{day}^{-1}$ and $f_2 = 0.00119 \, \text{days}^{-1}$, corresponding to periods of ~2,100 and 840 days, respectively.

## 5 Discussion and conclusions

It is widely accepted that the seismic cycle (or "seismic system") comprises six main stages (Figure 13) (Derode et al., 2021; Akopian and Kocharian, 2014). The stages are: (1) Over decades or years, small and medium asperities break continuously, resulting in a uniform rate of seismicity. (2) Asperities become locked, resulting in stress accumulation and decreasing seismic activity. (3) Weeks or days before a mainshock, important asperities progressively break along some sections (i.e., the foreshock stage). (4) Over a scale of hours, accumulated stresses overcome friction and blockages in the main asperities, causing the largest magnitude earthquake of the cycle. (5) Stress relaxation occurs after the mainshock and is characterised by numerous aftershocks of smaller magnitude over several weeks or months; this ceases when new asperities become locked. (6) Finally, the system returns to the initial, long-term, state.

In this paper, we have visualized that this mechanical description of the seismic cycle has an energetic analogy in terms of statistical physics and the Second Law of Thermodynamics. As argued in detail by De Santis *et al.* (2019), an earthquake can be considered as a phase transition, where continuous reorganization of stresses and forces reflects an evolution from equilibrium to non-equilibrium states. Therefore, entropy, which measures the number of accessible states for the present conditions of the systems, can be used as an indicator of the evolution of the system (e.g., (Telesca *et al.*, 2004, Vogel *et al.*, 2020). Stages 1–3 correspond to increasing stresses and the accumulation of seismic energy. During this inter-seismic period, the magnitudes of earthquakes are relatively uniform (or 'ordered') and entropy is relatively low. When a large earthquake occurs (stage 4), the rupture process triggers earthquakes with magnitudes of all sizes in a chaotic way, evolving to new conditions reaching a wider range of microstates in a disordered way, and the entropy increases. Finally, during the post-seismic state (stages 5 and 6), the system progressively recovers conditions similar to the initial ones.

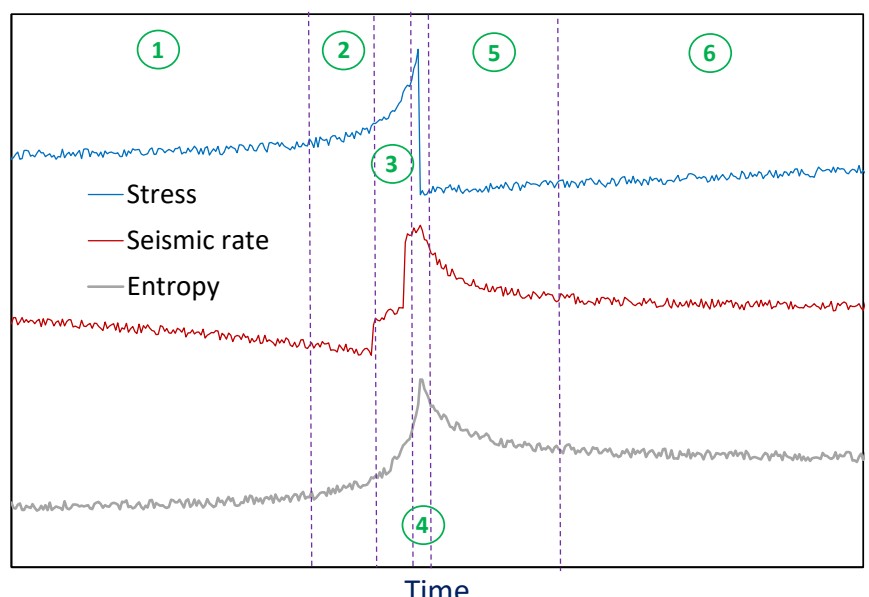

**Figure 13.** Seismic cycle from a mechanical perspective (i.e., stresses and seismic rate, which are shown in blue and red, respectively) and from a thermodynamic perspective (i.e., entropy, H, which is shown in grey). (1) Stage 1, the interseismic period, is characterised by approximately constant stress, seismic rate, and H. (2) Stage 2, the accumulation period, is characterised by modest increases in stress and H, but a modest decrease in seismic rate. (3) Stage 3, the foreshocks period, is characterised by increasing stress, seismic rate, and H. (4) Stage 4, the coseismic period, is characterised by an abrupt decrease in stress, but increases in the seismic rate and H. (5) Stage 5, the postseismic and aftershock period, is characterised by decreasing stress (i.e., relaxation), seismic rate, and H (towards the initial value). (6) Stage 6, during which the seismic cycle starts again.

Increasing entropy, *H*, from a thermodynamic perspective, is associated with an
irreversible transition from one state to another on both small (Scholz, 1968) and large
(e.g., Parsons *et al.*, 2008) scales. Using a high-quality catalogue of seismicity in northern
Chile, made possible owing to the IPOC network, we confirmed a strong temporal
correlation between entropy and the occurrence of earthquakes. Using the entropy value,
we could identify all earthquakes with magnitudes of > 6.5 in the catalogue. (i.e., seven
events from 2007 to 2014, with magnitudes ranging from 6.6 to 8.1)

However, it is important to note that changes in entropy are detected by analysing the
entire catalogue; that is, to detect a change in entropy associated with any event, data from
both before and after the event must be analysed. At present, this limits the use of this
method for seismic prediction. Further study is needed to determine a robust approach for
predicting how a time series will continue without prior knowledge; that is, to determine
threshold entropy values and trends that can be used to predict a significant event in the





immediate future. To achieve this, an absolute scale of entropy will be necessary. Earthquakes in zone A (0–80 km depth) tend to be tectonic in origin and have higher magnitudes than those in zones B and C (i.e., intermediate and deep earthquakes); as such,

they are of most concern from a risk management perspective. Our results show that the entropy changes associated with such events are much stronger when only data from this depth interval is considered; variations are of the order of one hundredth in zones B and C, but several tenths in zone A.

*Data availability.* The data are public and available at https://www.ipoc-network.org/data/ and in *Sippl et al. (2018)* available at http://doi.org/10.5880/GFZ.4.1.2018.001.

*Author contributions.* All authors contributed equally to the design of the methodology, discussion, analysis and revisions of the manuscript.

*Competing interests.* The authors declare no competing interests.

*Acknowledgements.* We would like to express our gratitude to the Integrated Plate Boundary Observatory Chile (IPOC) for collecting and sharing the data used in this work.

*Financial support.* This work was funded partially by the Spanish State Research Agency (SRA) under the grant PID2021-124701NB-C21 y C22., partially by the

FEDER/UAL Project UAL2020-RNM-B1980  and also partially by the research group RNM104 of the Junta de Andalucía. The University of Almeria funding for open access charge if applicable. EEV was supported by Fondecyt (grant number 1230055) and ANID through the Center for Development of Nanoscience and Nanotechnology (CEDENNA; grant number AFB220001).

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
