# Peer review of "Earthquake hazard characterization by using entropy: application to northern Chilean earthquakes"

_Natural Hazards and Earth System Sciences, 2023_

## Referee Comment (RC1)

The authors present an innovative relationship between the earthquake hazard and the Second Law of Thermodynamics by using the Shannon entropy, H, as a indicator of the changes in the seismic activity of a region.

The introduction is well structured, and it shows the evolution of the entropy concept from the very beginning to the current use in seismology as an indicator of the evolution of a system. The theoretical framework allows to understand the relationship between the entropy and the Gutenberg-Richter law although some steps should be explained better (see comments below). The methodology clearly explains how the variables needed to compute the entropy are obtained. The chosen datasets (northern Chilean seismicity) look adequate to probe the hypothesis although it should be explained better (see comments). The results are well presented and explained although some figures can be improved (see comments). The discussions are clear and meaningful.

The overall quality of the paper is very good, and the results can be interesting to the scientific community.

Specific questions/issues

1. The authors compute in the first part of the results the entropy for the whole catalogue (all depths). However, the number of earthquakes are not equally distributed for all the different depths and the seismic activity distribution is not the same for cortical and subduction or deep earthquakes so why is the reason of computing entropy using all of them if the physics is going to be different for the three depth regions?. The authors should explain why they do this better and if the idea is to demonstrate that the catalogue has to be spitted by depth region maybe even to carry out the fast fourier transform in Figure 6 may arrive to three main frequencies associated to the different entropy behaviour in each one of the three depth regions.
2. Additionally, in Figure 11 you compute the fast fourier transform only to the intermediate depth region. Why? You argue about the apparent periodicity of the entropic signal here but why it has to be periodic? And why only in the intermediate region? I would also see an apparent periodicity in the deep region with a period of about 1500 days. Why there is no periodicity for shallow earthquakes?. The stress loading rate is usually not uniform in time and a large earthquake may change the stress on the adjacent segment changing the seismic activity behaviour. Also the stress drop may change from event to event and the strength of the crust is not usually constant in time either. Therefore, the author should also try to address this.

3. How the uncertainty in magnitude and epicentral and depth location is taken into consideration during the analysis? The authors should also try to explain in the detail about this uncertainty and treatment in the catalogue and in the analysis.

Additionally, I would suggest you take into consideration the following corrections:

Paragraph 40. Sentence no.4. change *entropy is* by *entropy (S) is*

Equation (1) define in paragraph 45 what l and n means (you only have done it for k and $\Omega$)

Equation (4). As in this section you are still in speaking about entropy, I would suggest saying that assuming k=1 then it is possible to define the Shannon information entropy (H) combining equation (2) and (3). That will allow the reader to know exactly why H(p)=-I(p). Can you explain why k=1 is assumed? In this equation also you should explain why you have changed $\Omega$ to W (also in equation 5) and define W if it is what you want to write.

Equation (5) in the sentence where $\tau$ is a real number called the entropic index I suggest you improve the sentence saying where $\tau$ is named the entropic index and can, in principle, be any real number.

Equation (6) and (8). Sometimes you use x to represent the multiplication but others you do not use it, so I recommend removing the symbol x in all the equations.

Paragraph 95. In the sentence: *On the other hand, if we have N earthquakes and n denotes the number of earthquakes with magnitude M* you have to say *On the other hand, if we have N earthquakes and n denotes the number of earthquakes with magnitude equal to or larger than M* because it is needed to match with the Gutenberg-Richter relationship where n has that meaning.

Page 6. First sentence. After the words *the calculation of entropy* I would add between brackets a reference to the equation you are going to calculate)

Figure 1. It would be nice if you can add to the right of this figures two histograms (one with the magnitude frequency distribution and other with the depth frequency distribution)

Figure 8. Add dashed lines to the figure to separate the three mentioned regions.

Figure 10. I would remove this figure because it it the same information as in Figure 11 and Figure 11 is more illustrative than the previous one.

---

## Author Comment (AC1)

*The authors present an innovative relationship between the earthquake hazard and the Second Law of Thermodynamics by using the Shannon entropy, H, as a indicator of the changes in the seismic activity of a region.*

*The introduction is well structured, and it shows the evolution of the entropy concept from the very beginning to the current use in seismology as an indicator of the evolution of a system. The theoretical framework allows to understand the relationship between the entropy and the Gutenberg-Richter law although some steps should be explained better (see comments below). The methodology clearly explains how the variables needed to compute the entropy are obtained. The chosen datasets (northern Chilean seismicity) look adequate to probe the hypothesis although it should be explained better (see comments). The results are well presented and explained although some figures can be improved (see comments). The discussions are clear and meaningful.*

*The overall quality of the paper is very good, and the results can be interesting to the scientific community.*

First, we would like to thank referee number 1 for his/her valuable and constructive comments.

*Specific questions/issues*

*1. The authors compute in the first part of the results the entropy for the whole catalogue (all depths). However, the number of earthquakes are not equally distributed for all the different depths and the seismic activity distribution is not the same for cortical and subduction or deep earthquakes so why is the reason of computing entropy using all of them if the physics is going to be different for the three depth regions?. The authors should explain why they do this better and if the idea is to demonstrate that the catalogue has to be spitted by depth region maybe even to carry out the fast fourier transform in Figure 6 may arrive to three main frequencies associated to the different entropy behaviour in each one of the three depth regions.*

Referee number 1 is right in identifying different physical behaviours depending on the depth of the selected events. The behaviour of the shallow part of the crust (the relationship between stress, strain and fracture) is different from those occurring in the upper mantle. Our idea is to show how the methodology developed in the paper can identify large earthquakes even using the entire catalogue. This can be seen in figure 6 and it is necessary to emphasize that it is a first approximation to the study of the seismicity and the earthquake hazard characterization of the area. As can be seen in figure 6, other minor variations in entropy (concerning the Tocopilla and Iquique earthquakes) appear and this motivates us to analyse seismicity by depths as a second approximation.

To explain our approximation, we have added a new paragraph at the beginning of the Results section as referee # 1 has suggested.

*2. Additionally, in Figure 11 you compute the fast fourier transform only to the intermediate depth region. Why? You argue about the apparent periodicity of the entropic signal here but why it has to be periodic? And why only in the intermediate region? I would also see an apparent periodicity in the deep region with a period of about 1500 days. Why there is no periodicity for shallow earthquakes?. The stress loading rate is usually not uniform in time and a large earthquake may change the stress on the adjacent segment changing the seismic activity behaviour. Also the stress drop may change from event to event and the strength of the crust is not usually constant in time either. Therefore, the author should also try to address this.*

Indeed, the periodic behaviour could be in the three regions A, B and C. In fact, as referee #1 correctly points out, in region C a period of 1500 days can be appreciated, although, as the authors indicated, maybe it would be necessary a longer time series be able to conclude the existence of this periodicity (the "sample" only has 3000 days). In short, in region A there are very small and irregular periods, possibly associated with a stress-loading rate usually not uniform in time because, as is well known, the strength of the crust is not constant. What happens in the deeper regions, B and C, should be associated with the extremely complex mechanisms of subduction and slag detachment. However, the periodicity that we had better observe is that of region B and, for this reason, we have carried out the fast Fourier transform only to the intermediate depth region.

To explain better why only use the fast Fourier transform in region B, we have added a new paragraph at line 256 as referee # 1 has suggested.

*3. How the uncertainty in magnitude and epicentral and depth location is taken into consideration during the analysis? The authors should also try to explain in the detail about this uncertainty and treatment in the catalogue and in the analysis.*

IPOC catalogue used in our work is available from the important and recent paper by Sippl et al. (2018). In that work, cited in our manuscript, authors presented a regional earthquake catalogue containing 101,601 double-difference relocated earthquakes to show high-resolution seismicity images of the northern Chile subduction zone forearc. The dataset is extended to 8 years of continuous seismic waveform data using automatic event detection and phase-picking routines. The uncertainty in hypocentral location was computed by using the probabilistic routine NonLinLoc (Lomax et al., 2000); authors proved that location errors inside the network are typically small ($< 5$ km) and hypocentral depth errors appear to be systematically larger than horizontal location errors, although this effect is small inside the network. Following Sippl et al. (2018), local magnitudes were determined from maximum amplitudes on the horizontal components after Hutton and Boore (1987) and they retrieved a total of 1,200,404 P and 688,904 S phase picks with average uncertainties of 0.11 and 0.37 s, respectively. Definitively, IPOC catalogue is a high-quality database. All these details are included in the Sippl et al. (2018) paper and we refer to readers to that work (line 192 in our paper). In our opinion, the IPOC catalogue represented an effort to minimize sources of error. However, these data are not absolutely free of

error, which is not uniform in time or place. Our approach has been to begin with a general, kind of average approach and later illustrate differences with respect to depth, which is probably the less precise parameter since it is strongly model dependent especially for deeper seisms. The latitude and longitude coordinates are more precise for shallow earthquakes. Errors in magnitude can be of limited character only, since they handle 2 significant figures at the most. Time determination is almost error free, since it is determined from the arrival of the P waves to several stations. The kind of errors present in these problems are known and they should be bore in mind when analysing the present and other results.

Hutton, L. K., & Boore, D. M. (1987). The Ml Scale in Southern California. Bulletin Of The Seismological Society Of America, 77(6), 2074–2094.

Lomax, A., Virieux, J., Volant, P., & Berge-thierry, C. (2000). Chapter 5 Probabilistic earthquake location in 3D and layered models. In A. Lomax, et al. (Eds.), Advances in Seismic Event Location (pp. 101–134). Amsterdam: Kluwer Academic Publishers.

Sippl, C., Schurr, B., Asch, G., Kummerow, J., Seismicity structure of the northern Chile forearc from >100,000 double-difference relocated hypocenters, J. Geophys. Res. Solid Earth 123, 4063–4087; doi: 10.1002/2017JB015384 (2018).

***Additionally, I would suggest you take into consideration the following corrections:***

***Paragraph 40. Sentence no.4. change entropy is by entropy (S) is***

Done

***Equation (1) define in paragraph 45 what l and n means (you only have done it for k and W)***

We are sorry. "l" and "n" must be joined as "ln" , the natural logarithm; we have corrected it.

***Equation (4). As in this section you are still in speaking about entropy, I would suggest saying that assuming k=1 then it is possible to define the Shannon information entropy (H) combining equation (2) and (3). That will allow the reader to know exactly why H(p)=-I(p). Can you explain why k=1 is assumed? In this equation also you should explain why you have changed W to W (also in equation 5) and define W if it is what you want to write.***

We believe that your suggestion is right and we have rewritten the paragraph following your recommendation. Moreover, a new reference was added (e.g., Truffet, 2018) to better explain why $k = 1$. Finally, we have unified the symbols W and $\Omega$.

L. Truffet, Shannon Entropy Reinterpreted, Reports on Mathematical Physics, 81, 3, 2018, 303-319, https://doi.org/10.1016/S0034-4877(18)30050-8.

*Equation (5) in the sentence where t is a real number called the entropic index I suggest you improve the sentence saying where t is named the entropic index and can, in principle, be any real number.*

This suggestion have been incorporated in the manuscript

*Equation (6) and (8). Sometimes you use x to represent the multiplication but others you do not use it, so I recommend removing the symbol x in all the equations.*

Of course, you are right and the "x" symbol has been removed.

*Paragraph 95. In the sentence: On the other hand, if we have N earthquakes and n denotes the number of earthquakes with magnitude M you have to say On the other hand, if we have N earthquakes and n denotes the number of earthquakes with magnitude equal to or larger than M because it is needed to match with the Gutenberg-Richter relationship where n has that meaning.*

Done.

*Page 6. First sentence. After the words the calculation of entropy I would add between brackets a reference to the equation you are going to calculate)*

Done.

*Figure 1. It would be nice if you can add to the right of this figures two histograms (one with the magnitude frequency distribution and other with the depth frequency distribution)*

Done. Figure 4 and figure 8 were added to figure 1.

*Figure 8. Add dashed lines to the figure to separate the three mentioned regions.*

Done

*Figure 10. I would remove this figure because it it the same information as in Figure 11 and Figure 11 is more illustrative than the previous one*

Done

---

## Author Comment (AC2)

*In the paper "Earthquake hazard characterization by using entropy: application to northern Chilean earthquakes" the authors discuss a statistical physics approach for the characterization of seismicity evolution and dynamics in Northern Chile. To this end, the authors use the relationship between the Gutenberg-Richter scaling relation and the Shannon entropy to demonstrate variations in entropy associated with the occurrence of strong earthquakes in this region. The manuscript is generally well-written and organized, the methodology is sound, and the results present some interest for the scientific community. Therefore, I recommend its publication after some minor revisions listed below.*

First, we would like to thank referee number 2 for his/her valuable and constructive comments.

*1) In Equation 6 and so on, the upper limit of the integral, representing the interval of earthquake magnitudes, is infinity. However, there is a maximum magnitude up to which earthquakes occur. I would suggest substituting the infinity symbol with Mmax.*

Thank you very much for this suggestion. Of course, it has been done.

*2) In Page 4, the annotation given first to the parameter n is "the number of earthquakes with magnitude M", whereas later on "the cumulative number of earthquakes with a magnitude equal to or larger than M". The annotation given to particular parameters should be consistent throughout the text.*

You have right and we have corrected this mistake to have coherence in the whole text.

*3) Equation 17 has also been derived by De Santis et al. (2011). Provide the appropriate references and/or discussion.*

Done. Of course, we have added De Santis et al. (2011).

*4) The authors use the MAXC method to estimate the magnitude of completeness (Mc) of their catalog. Woessner and Wiemer (BSSA, 2005) suggested that Mc calculated with this method should be corrected to +0.2 units of magnitude to give more robust estimation of the b-value. Did the authors consider this correction?*

Referee #2 is right in his/her appreciation: it is well known that MAXC method generally underestimated $M_C$ value. In their paper, Woessner and Wiemer (2005) state that:" *The application of the EMR and MAXC approaches to the 1992 Landers aftershock sequence shows that Mc was slightly underestimated by 0.2 in Wiemer and Katsumata (1999)*". And, finally, their conclusions indicated: *"...for a fast analysis of Mc, we recommend using the MAXC approach in combination with the bootstrap and add a correction value (e.g., $M_C = M_C(MAXC) + 0.2\ Mc$)*".

However, when the number of earthquakes to be considered is important from a statistical point of view, the best option is the MAXC technique. Thus, e.g. De Santis *et al.* (2011) stated that: "*The choice for this value of $M_0$ 1.4 was made by inspecting the magnitude frequency and cumulative distributions*

*over the period of concern to check the catalogue completeness. We recognize that this choice of $M_0$ is a little lower than the value given for the same region by a recent evaluation of the spatio-temporal behaviour of $M_0$ of the same catalogue over Italy. (…) The dense distribution of the more recent seismic network (…) and the careful check by the personnel dedicated to the operations of seismic event detection support the value here proposed for $M_0$. In addition, our choice of $M_0$ 1.4 allows us to use a greater number of events than those eventually obtained considering a greater magnitude threshold, thus improving the statistics of our analysis*". ($M_0$ in their paper refer to $M_C$).

As in the case of the De Santis et al. (2011) work, the IPOC catalogue used recordings from the IPOC seismic network (GFZ & CNRS-INSU) as well as auxiliary permanent or temporary stations that were deployed in the years 2007–2014; moreover, permanent stations from the CSN (Centro Sismológico Nacional) and GEOFON (GEOFON Data Center, 1993), WestFissure network operated by the Free University of Berlin, and the MINAS and IQ networks operated by GFZ Potsdam were used. On the other hand, scientific personnel working on the IPOC network include the GFZ German Research Centre for Geosciences, Potsdam Germany; the Centre National de la Récherche Scientifíque Paris (C.N.R.S.), France; the Centro Sismológico Nacional, Chile; the Ecole Normale Supérieure, Paris, France; the Freie Universität Berlin, Germany; the GEOMAR Helmholtz Centre for Ocean Research Kiel, Germany; the Institut de Physique du Globe Paris (IPGP), France; the Pontificia Universidad Católica de Chile, Santiago, Chile; the Universidad Católica del Norte, Antofagasta, Chile and finally the Universidad de Chile, Santiago, Chile.

Although the correction of MC(MAX) + 0.2 is correct, our interest in selecting the maximum possible number of earthquakes (and their high quality), led us to slightly underestimate the value of MC.

**5) As the authors discuss in Figure 9, the threshold magnitude (Mc in my previous comment) varies with depth. However, in their analysis of the entire catalog, they use a common threshold magnitude for all depth ranges. In addition, it is possible that Mc also varies with time, and it should be estimated in the temporal windows. The proper estimation of Mc (or $M_0$) is crucial for the determination of the b-value (see Eq. 18).**

In spite of the possible depth and time variations of the GR parameters we have preferred to do just one consolidated analysis with richer statistics, representing an average behaviour of the distribution of the > 100,000 earthquakes included in this study.

**6) In Figure 1 show the position of the second largest region on the globe.**

Done

**7) *The authors mention the Gutenberg-Richter scaling relation in Fig.4, as well as in other figures (Fig.9). However, in these figures only the cumulative and discrete frequency-magnitude distribution is shown. Show also the Gutenberg-Richter relation and the associated a and b parameters.***

Done

**8) *What do the colors indicate in Fig.6?***

Usually, but it is not universal in all countries, the earthquake hazard colour code set up that "cool" colours, such as green or blue, are related with not dangerous earthquakes, whereas "hot" colour, such as orange or red, are related with earthquakes which higher magnitudes and then, the potential seismic danger is associated to them. Nevertheless, referee #2 is right and the colour could confuse the reader; therefore, we remove colour scale and now, the graph is black. For coherence, figure 3 is also redrawn in black colour.

---

## Author Comment (AC4)

*In the paper "Earthquake hazard characterization by using entropy: application to northern Chilean earthquakes" the authors discuss a statistical physics approach for the characterization of seismicity evolution and dynamics in Northern Chile. To this end, the authors use the relationship between the Gutenberg-Richter scaling relation and the Shannon entropy to demonstrate variations in entropy associated with the occurrence of strong earthquakes in this region. The manuscript is generally well-written and organized, the methodology is sound, and the results present some interest for the scientific community. Therefore, I recommend its publication after some minor revisions listed below.*

First, we would like to thank referee number 2 for his/her valuable and constructive comments.

*1) In Equation 6 and so on, the upper limit of the integral, representing the interval of earthquake magnitudes, is infinity. However, there is a maximum magnitude up to which earthquakes occur. I would suggest substituting the infinity symbol with Mmax.*

Thank you very much for this suggestion. Of course, it has been done.

*2) In Page 4, the annotation given first to the parameter n is "the number of earthquakes with magnitude M", whereas later on "the cumulative number of earthquakes with a magnitude equal to or larger than M". The annotation given to particular parameters should be consistent throughout the text.*

You have right and we have corrected this mistake to have coherence in the whole text.

*3) Equation 17 has also been derived by De Santis et al. (2011). Provide the appropriate references and/or discussion.*

Done. Of course, we have added De Santis et al. (2011).

*4) The authors use the MAXC method to estimate the magnitude of completeness (Mc) of their catalog. Woessner and Wiemer (BSSA, 2005) suggested that Mc calculated with this method should be corrected to +0.2 units of magnitude to give more robust estimation of the b-value. Did the authors consider this correction?*

Referee #2 is right in his/her appreciation: it is well known that MAXC method generally underestimated $M_C$ value. In their paper, Woessner and Wiemer (2005) state that:" *The application of the EMR and MAXC approaches to the 1992 Landers aftershock sequence shows that Mc was slightly underestimated by 0.2 in Wiemer and Katsumata (1999)*". And, finally, their conclusions indicated: *"...for a fast analysis of Mc, we recommend using the MAXC approach in combination with the bootstrap and add a correction value (e.g., $M_C = M_C(MAXC) + 0.2\ Mc$)"*.

However, when the number of earthquakes to be considered is important from a statistical point of view, the best option is the MAXC technique. Thus, e.g. De Santis *et al.* (2011) stated that: "*The choice for this value of $M_0$ 1.4 was made by inspecting the magnitude frequency and cumulative distributions*

*over the period of concern to check the catalogue completeness. We recognize that this choice of $M_0$ is a little lower than the value given for the same region by a recent evaluation of the spatio-temporal behaviour of $M_0$ of the same catalogue over Italy. (…) The dense distribution of the more recent seismic network (…) and the careful check by the personnel dedicated to the operations of seismic event detection support the value here proposed for $M_0$. In addition, our choice of $M_0$ 1.4 allows us to use a greater number of events than those eventually obtained considering a greater magnitude threshold, thus improving the statistics of our analysis*". ($M_0$ in their paper refer to $M_C$).

As in the case of the De Santis et al. (2011) work, the IPOC catalogue used recordings from the IPOC seismic network (GFZ & CNRS-INSU) as well as auxiliary permanent or temporary stations that were deployed in the years 2007–2014; moreover, permanent stations from the CSN (Centro Sismológico Nacional) and GEOFON (GEOFON Data Center, 1993), WestFissure network operated by the Free University of Berlin, and the MINAS and IQ networks operated by GFZ Potsdam were used. On the other hand, scientific personnel working on the IPOC network include the GFZ German Research Centre for Geosciences, Potsdam Germany; the Centre National de la Récherche Scientífíque Paris (C.N.R.S.), France; the Centro Sismológico Nacional, Chile; the Ecole Normale Supérieure, Paris, France; the Freie Universität Berlin, Germany; the GEOMAR Helmholtz Centre for Ocean Research Kiel, Germany; the Institut de Physique du Globe Paris (IPGP), France; the Pontificia Universidad Católica de Chile, Santiago, Chile; the  Universidad Católica del Norte, Antofagasta, Chile and finally the Universidad de Chile, Santiago, Chile.

Although the correction of MC(MAX) + 0.2 is correct, our interest in selecting the maximum possible number of earthquakes (and their high quality), led us to slightly underestimate the value of MC.

**5) As the authors discuss in Figure 9, the threshold magnitude (Mc in my previous comment) varies with depth. However, in their analysis of the entire catalog, they use a common threshold magnitude for all depth ranges. In addition, it is possible that Mc also varies with time, and it should be estimated in the temporal windows. The proper estimation of Mc (or $M_0$) is crucial for the determination of the b-value (see Eq. 18).**

In spite of the possible depth and time variations of the GR parameters we have preferred to do just one consolidated analysis with richer statistics, representing an average behaviour of the distribution of the > 100,000 earthquakes included in this study.

**6) In Figure 1 show the position of the second largest region on the globe.**

Done

*7) The authors mention the Gutenberg-Richter scaling relation in Fig.4, as well as in other figures (Fig.9). However, in these figures only the cumulative and discrete frequency-magnitude distribution is shown. Show also the Gutenberg-Richter relation and the associated a and b parameters.*

Done

*8) What do the colors indicate in Fig.6?*

Usually, but it is not universal in all countries, the earthquake hazard colour code set up that "cool" colours, such as green or blue, are related with not dangerous earthquakes, whereas "hot" colour, such as orange or red, are related with earthquakes which higher magnitudes and then, the potential seismic danger is associated to them. Nevertheless, referee #2 is right and the colour could confuse the reader; therefore, we remove colour scale and now, the graph is black. For coherence, figure 3 is also redrawn in black colour.

[revised manuscript text omitted]

**Comentado [UdW15]:** Correction #8 and #9 from referee #1. Figure 1 include the magnitude frequency distribution (figure 4 in the original manuscript) and the depth frequency distribution (figure 8 in the original manuscript). Moreover, in figure 1c, we have added dashed lines to separate the three mentioned regions,

Correction #6 from referee #2. We show the position of the largest region on the globe. In addition, in figure 1b we show the Gutenberg-Richter relation and the associated a and b parameters.

**Comentado [UdW16]:** Question/issue #1 from referee #1

We added this paragraph to better explain our results, as Referee # 1 suggested.

[revised manuscript text omitted]

**Comentado [b19]:** Question/issue #2 from referee #1

We added this paragraph to better explain our results, as Referee # 1 suggested.

[Figure]

**Figure 7.** Epicentrally represented earthquake activity and non-cumulative and cumulative Gutenberg–Richter relationships in zones A–C for earthquakes with magnitudes of $> 3.0$. (a) Zone A (0–80 km), (b), zone C (80–160 km), and (c) zone C (>160 km). Symbol colours denote earthquake magnitude: yellow circles = 3.0–3.9, cyan circles = 4.0–4.9, blue circles = 5.0–5.9, green triangles = 6.0–6.9, and red stars = $> 7.0$. Based on the maximum curvature (MAXC) technique (Wiemer and Wyss, 2000), $M_0$= 2.2 in zones $A$ and B, and 3.2 in zone C.

**Comentado [UdW20]:** Correction #7 from referee #2. Gutenberg-Richter relation and the associated a and b parameters is shown in the figure.

[revised manuscript text omitted]